# Implementation of a Psychiatric Consultation for Healthcare Workers during First Wave of COVID-19 Outbreak

**DOI:** 10.3390/ijerph19084780

**Published:** 2022-04-14

**Authors:** Lamyae Benzakour, Gérard Langlois, Verena Marini, Alexandra Groz, Chiara Chiabotto, Diana Apetrei, Bruno Corneau, Guido Bondolfi

**Affiliations:** 1Liaison Psychiatry and Intervention Crisis Service, Geneva University Hospital, 1205 Geneva, Switzerland; chiara.chiabotto@hcuge.ch (C.C.); diana.apetrei@hcuge.ch (D.A.); guido.bondolfi@hcuge.ch (G.B.); 2Health Care Directorate, Geneva University Hospital, 1205 Geneva, Switzerland; gerard.langlois@hcuge.ch (G.L.); verena.marini@hcuge.ch (V.M.); alexandra.groz@hcuge.ch (A.G.); bruno.corneau@hcuge.ch (B.C.)

**Keywords:** healthcare workers, COVID-19, psychiatric consultation, burnout, Post Traumatic Stress Disorder (PTSD), stress

## Abstract

Background: Prevention and management strategies of mental suffering in healthcare workers appeared as important challenges during the COVID-19 pandemic. This article aims to: (1) show how potential psychiatric disorders for healthcare workers (HCW) during the first wave of the COVID-19 outbreak were identified; (2) present an activity report of this consultation; and (3) analyze and learn from this experience for the future. Methods: We performed a retrospective quantitative analysis of socio-demographic and clinical data, in addition to psychiatric scales scores for the main potential psychiatric risks (PDI, PDEQ, PCL-5, HADS, MBI-HSS) and post-hoc qualitative analysis of written interviews. Results: Twenty-five healthcare workers consulted between 19 March 2020 and 12 June 2020. We found 78.57% presented high peritraumatic dissociation and peritraumatic distress, 68.75% had severe anxiety symptoms, and 31.25% had severe depression symptoms. Concerning burnout, we found that 23.53% had a high level of emotional exhaustion. In the qualitative analysis of the written interview, we found a direct link between stress and the COVID-19 pandemic, primarily concerning traumatic stressors, and secondarily with work-related stress. Conclusions: Early detection of traumatic reactions, valorization of individual effort, and limitations on work overload appear like potential key preventive measures to prevent psychiatric complications for healthcare workers in the context of the COVID-19 pandemic.

## 1. Introduction

The outbreak of coronavirus disease-19 (COVID-19) emerged in December 2019 in Wuhan (China), and has so far consisted of three waves which have already given rise in the world to 162,500,000 confirmed cases and has killed 3,369,259, as of May 2021 [1]. To date, in Switzerland, 674,138 cases have been laboratory-confirmed, equating to 7833/100,000 inhabitants, and 10,080 deaths (117.13/100,000 inhabitants) caused by COVID-19 have been recorded [2]. From the pandemic’s onset, healthcare workers (HCW) have needed to adapt to this unprecedented situation to avoid hospital saturation, and limit both deaths and severe complications. For HCW, uncertainty about the length of the pandemic, the need to adapt to new care management due to the outbreak, and the lack of knowledge about COVID-19 were the most prominent stress factors. These challenges were identified early-on as risk factors of psychological suffering for hospital workers [3]. During the initial lockdown period from 16 March to 19 April 2020, HCW were confronted by the virus in the name of the collective good and quickly became, by force of circumstance, the “soldiers on the frontline”. In many countries, this image of HCW was solidified by media and societal rituals, manifested by applause given at the end of each day. Within this context, it may have been difficult for HCW to recognize and acknowledge their psychological suffering.

The first published studies on psychological suffering of HCW during the COVID-19 pandemic confirmed the existence of this heavy psychological burden, especially for anxiety, depression, and insomnia [4,5]. The influence on mental symptoms of age and gender were considered as how the HCW presented itself at the consultation and not the biological attribute according to Sex and Gender Equity in Research—SAGER, occupation, specialization, the types of activity performed, and their proximity to COVID-19 patients were also highlighted [4,5]. Moreover, post-traumatic stress disorder (PTSD) risk in HCW during previous coronavirus epidemics was highlighted, as well as during the COVID-19 pandemic [6].

Certain variables were found to be particularly relevant as PTSD risk factors, such as the female gender, older age, exposure level, working role, years of work experience, social and work support, job organization, presence of quarantine, marital status, and resilience factors, such as coping styles and social support [6].

Prior to implementing a consultation facility (CovidPsy) for HCW, we attempted to identify potential mental health risks to design our consultation methodology and adapt the terms of care that could be offered. Data from previous respiratory pandemics of severe acute respiratory syndrome (SARS) in 2002–2003, and Middle East respiratory syndrome (MERS) in 2012 were used, as well as information from foreign media outlets, such as those in China where the pandemic began. Then, these data were adapted to the local context before implementing the CovidPsy consultation. We were aware that the experiences and the management of the COVID-19 crisis varied greatly from one country to another, but at this period, we had no published studies concerning the impact of COVID-19 pandemics on mental health for HCW, and no available data for Switzerland or a comparable country, so we decided to consider the available data and to consider the potential psychiatric issues that were previously described even in different contexts of crisis and to think about potential psychiatric issues knowing the specificities of the COVID-19 pandemics as explained above.

We distinguished two main categories of stress factors: a) work-related stress during COVID-19 outbreak; and b) direct stress consequences of COVID-19.

Faced with an uncontrollable viral outbreak and its treatment, HCW might have felt powerless to help their patients. For example, as the influx of patients increased, urgent improvised decisions had to be made to spare care resources, which at the time were almost entirely dedicated to managing COVID-19, and specifically implementing algorithms to prioritize care. As a result, the clinical activity of HCW was brutally and rapidly transformed, shaking the very basis of their professional identity. For many, time stood still from the start of this first wave of the health crisis. Unpredictability contaminated all aspects of the HCW’s daily life, and especially when exposed to certain COVID-19 patients who experienced rapid deterioration.

In our university hospital, HCW were “requisitioned”, meaning that their vacation was suspended for an indefinite period of time, and were given rare instances of time to decompress. Faced with difficulties anticipating healthcare resource needs during this unprecedented crisis, as well as absenteeism linked to staff contamination by SARS-CoV-2, HCW had to change medical units accordingly, regularly modifying their schedule, thus creating very unstable and intense working conditions.

In light of the pandemic, non-COVID-19 clinical activities had to be abruptly suspended, giving the impression of patient abandonment. The issue of patient triage arose, on the grounds of efficiency: COVID-19 patients who needed to be given priority in intensive care units were those for whom an improvement in prognosis was considered more likely. Therefore, the management priority was for COVID-19, to the detriment of other healthcare activities and their consequences [7]. In some instances, these situations led to value conflicts and a profound sense of loss in many HCW who no longer felt useful in the exercise of their profession.

Finally, during the first wave of the COVID-19 outbreak, concomitant losses of control and sense, as well as workload, increased, which are recognized as important factors of burnout, as described by Maslach [8]. HCW are known to be exposed at higher risk of burnout and their complications [9], and some work factors such as lack of input or control for physicians, excessive workloads, inefficient work processes, clerical burdens, have been identified for physicians [10]. For these reasons, we considered it was important to investigate burnout and treat it in the context of this psychiatric consultation for HCW.

In the course of their work, HCW were exposed to potential contamination by SARS-CoV-2 and subsequent increased risk of transmission to loved ones. The contradictory information surrounding the subject contributed to the caregivers’ insecurity, as it was objectively impossible to guarantee protection, due to a lack of knowledge on the transmission modes of the virus.

Specifically, HCW could find themselves faced with unpredictable, sudden, and numerous deaths, quantities they were not accustomed to, including those in intensive care units, even if they had previous experience with critical situations. In their personal lives, like the rest of the general population, many HCW were also impacted due to the illness and deaths of those closest to them.

The confluence for many HCW of significant stress, the prioritization of professional duties above all others, and the restriction of social contacts and leisure time outside the hospital, has contributed to increases in the risk for psychological distress such as anxiety, depression and traumatic experience during this period. Furthermore, holidays for hospital staff were abolished in most countries. In some cases, HCW were beginning to report situations of very painful rejection from their relatives for fear of contamination.

Faced with a highly stressful situation, the subject’s coping skills may be overwhelmed and give rise to reactive depressive or anxious symptoms. Depending on the level of personal resources available to mobilize coping strategies, and confronted with the same situation, some workers will develop anxious and/or depressive symptoms, whereas others may be able to adapt. Interindividual variability in reactions to a very stressful and unprecedented situation such as the COVID-19 pandemic was expected.

This COVID-19 context created situations of potential or actual death of patients, which meet the definition of traumatic events, as defined in the Diagnostic and Statistical Manual of Mental Disorders (DSM–5) [11]. Subjects who had a history of trauma, either related or not to their professional activity, and/or a psychiatric history of depression, had a greater vulnerability for the risk of acute stress disorder (ASD) or post-traumatic stress disorder (PTSD) [12]. Importantly, PTSD risk factors have been well studied during previous coronavirus epidemics [13]. On the basis of past experiences and the identification of risk factors directly due to the COVID-19, a risk of PTSD in HCW seemed to be important and warranted further investigation. PTSD is associated with a poor prognosis and an important risk of comorbidities such as substance and alcohol abuse [14]. Prevention strategies are known to reduce the risk of chronic evolution to PTSD, such as an early identification and care for people at risk for PTSD, even if the effectiveness of some interventions like cognitive-behavioral therapies (CBT), eye movement desensitization reprocessing (EMDR), and pharmacological strategies require further study [15,16].

Important emphasis is placed on the potential psychiatric complications of this health crisis. However, in traumatic situations, psychological benefits grouped under the emerging notion of post-traumatic growth can, on the contrary, develop in the aftermath of a traumatic event. Authors of a qualitative review on disaster-exposed organizations identified several protective factors after a disaster: training, experience, and perceived (personal) competence; social support; and effective coping strategies. Post-traumatic growth can provide a greater appreciation of life and relationships, enhancing self-esteem and providing a sense of accomplishment and better understanding of an individual’s work [17]. The exploration of these protective factors appeared as important regarding the risk factors for PTSD in the time of COVID-19.

At the Geneva University hospitals, in the beginning of the first wave of COVID-19, we implemented several strategies in order to prevent and manage early psychological suffering among HCW. Psychologists in COVID-19 units and hypnosis sessions were deployed. The service of liaison psychiatry, the staff health service and the Health Care Directorate received an official mandate on 16 March 2020 from the medical director of the University Hospital of Geneva, and three days later, on 19 March 2020, a psychiatric consultation was offered to the hospital workers. Our paper aims to (1) show how potential psychiatric disorders for HCW during the first wave of the COVID-19 outbreak were identified; (2) present an activity report of this consultation; and (3) analyze and learn from this experience for the future.

## 2. Materials and Methods

### 2.1. Participants

All HCW (clinical and non-clinical HCW) of our university hospital were able to ask for a consultation at the permanence without an appointment, not only employees who were in charge of COVID-19 patients. We have counted that 25 HCW consulted, and that we provided 52 consultations from the 16 March to the 12 June 2020. Of the 25 employees, only 18 gave informed consent, which allowed us to retrospectively analyze their personal and clinical data for the study. The mean age was 40, 67 years old (25–58 years old), with a majority of women (14; 77.78%). We found that 9 nurses, 2 physicians, and 2 medical students that were requisitioned in COVID-19 wards during the first wave, as well as 4 other clinical HCW, and 1 administrative hospital employee, consulted. We found 72.22% (*n* = 13) of HCW consulting at the permanence were frontline health care workers, meaning those who interacted directly with COVID-19-positive, or potentially positive, patients, 77.78% were women (*n* = 14), and 83.33% married or living in common law (*n* = 15).

### 2.2. Interventions

Recommendations for setting up support systems for caregivers were quickly disseminated by the World Health Organization at the start of the crisis [3,18] based on previous epidemics, highlighting the need to organize the system for the prevention and the management of the mental suffering of the HCW. Within a few days in our hospital, the CovidPsy psychiatric consultation service was not only established, but support psychologist positions in the COVID-19 wards hypnosis sessions and a hotline were deployed. Material aids were offered, such as parking spaces, accommodation in hotels for people who lived far away, and free meals.

After the first step, when we received an official mandate, we made a request to two consultation offices. We implemented a 7 day a week, 9 AM to 6 PM consultation service, to receive any HCW requesting help, free of charge and without an appointment. We chose a name and conceived of a framework for the consultation.

The psychiatric consultation team was composed of hospital and private psychiatrists-psychotherapists, and clinical specialist nurses in psychiatry, whose usual clinical activity was reduced, thanks to the solidarity of other psychiatric services and the Health Care Directorate.The final team for the psychiatric consultation was composed of 9 psychiatrists-psychotherapists and 8 clinical specialist nurses in psychiatry to ensure the presence of at least one psychiatrist-psychotherapist and nurse at all times during opening hours. The psychiatric intervention policies (guidelines for the intervention, organization of the permanence, and establishment of the duty schedule) were defined. Our crisis intervention and algorithm models were inspired by disaster psychiatry [19]. They consisted of a preventive model based on the identification of traumatic stressors and high-risk subjects of psychological suffering [12].

Based on data and the knowledge from previous epidemics of how mental health is impacted in hospital workers, we identified the following risks: (1) burnout; (2) trauma disorders (acute stress disorder trauma, vicarious trauma, post-trauma stress disorder); and (3) anxiety and depression symptoms. We proposed a systematic screening of these risks at the beginning of the consultation using the French version of The Pocket Guide to the DSM-5^TM^ Diagnostic Exam, whose license has been obtained for each survey [20] and completed the evaluation with some questions on psychiatric history, their family and social circles, and working conditions. Depending on this evaluation, this was followed by a personalized therapeutic intervention using specific guidelines. Consultations were carried out by a psychiatrist-psychotherapist and a clinical specialist nurse in psychiatry to encourage complementary interventions and to partition emotional burden in a face-to-face session. If acute stress symptoms were identified, recommended interventions after a traumatic event such as defusing intervention, psychoeducation intervention on PTSD, and/or eye movement desensitization reprocessing (EMDR) for recent trauma were provided. If burnout symptoms were identified, we gave feedback about these symptoms to the hospital workers directly and suggested a sick leave. In front of anxiety symptoms and/or acute stress symptoms, we used stress management tools like safe place, cardiac coherence and mindfulness interventions. For all the clinical situations, analyses of stress factors at work were conducted with the person who consulted and a search of strategies to cope with them was undertaken. Personal resources were sought and reinforced as much as possible. Medications could also be prescribed depending on the psychiatric evaluation. We proposed short interventions which should not exceed three consultations, with a few exceptions. Indeed, we considered that if the collaborators required a longer intervention, that their follow-up should be able to be continued outside of this permanence and referred them. This was according to emergency and the short interventions model, and because of a lack of availability from the team to make longer follow-ups. We organized a referral for another follow-up if required because the HCW was not clinically sufficiently improved after three sessions at our permanence, or if the HCW asked for other type of follow-up (private psychiatrist-psychotherapist or psychologist-psychotherapist or consultation center depending on the hospital). Anyway, HCW knew that they could contact us at any moment after the intervention if they need via the hotline, but we did not provide any systematic evaluation after the intervention that was not built in a research context but in a clinical goal.

The number of sessions depended on the clinical assessment and the therapeutic goals that we defined with the HCW. The number of sessions for one HCW varied from 1 to 11 sessions (µ = 2.7), and the duration by session varied from 45 to 150 min (µ = 89 min).

### 2.3. Materials

We collected personal data (data birth, phone number, marital status, phone number, mail), information on working conditions (position held, department, work in COVID-19 unit, change of service due to COVID-19 outbreak, …), medical and psychiatric history and previous trauma, risk factors for severe forms of COVID-19, and contamination by SARS-CoV-2. We chose to use a systematic screening with scales to look for these risks before each consultation to adapt the intervention to the needs of the worker. HCW completed different validated tools in their French versions to look for main psychiatric situations that were expected, before the intervention (methods previously described): (1) The Maslach Burnout Inventory-Human Services Survey (MBI-HSS) was used in this survey by obtaining a license (see Appendix A), and consisted of three dimensions: emotional exhaustion (EE), depersonalization (DP), and personal achievement (PA). The level of burnout was considered high if EE was ≥27, DP was ≥13, and PA was ≤21; moderate if EE was 17–26, DP was 7–12, and PA was 38–22; and low if EE was ≤16, DP was ≤6, and PA was ≥39 [21,22]. (2) The Hospital Anxiety and Depression Scale (HADS) was also used, which assesses transdiagnostic symptoms of anxiety and depression in patients with a somatic disorder, using a cutoff total score of 11 for anxiety and for depression [23]. (3) The peritraumatic distress inventory (PDI), which screens for distress symptoms during and immediately following a traumatic event, using a cutoff at 15 to identify a high risk of future PTSD, was used [24,25]; and also (4) the Peritraumatic Dissociative Experiences Questionnaire (PDEQ) which screens for dissociative symptoms such as depersonalization and derealization during and immediately following a traumatic event, using a cutoff at 15 to identify a high risk of future PTSD [26,27]. (5) The Posttraumatic Stress Disorder Checklist for DSM-5 (PCL-5) was also used, which assesses current symptoms of PTSD, using a cutoff score of 33 to identify a PTSD diagnosis which was given instead of PDI and PDEQ, only in the case of PTSD diagnosis made in the presence of PTSD criteria present more than one month according to the DSM5 [20], to assess PTSD severity [28,29].

### 2.4. Data Analysis

#### 2.4.1. Quantitative Analysis

Descriptive statistical analyses were made using Excel^®^ and an R^®^ software package provided by the R Foundation for Statistical Computing. Qualitative variables were expressed as frequencies and percentages. Quantitative variables were expressed as means with minimum and maximum values.

#### 2.4.2. Qualitative Analysis

We also performed a qualitative analysis of the 47 written interview notes of HCW that consented to participate in the study: one of the researchers of the team systematically analyzed the semi-structured interview notes written by clinicians of the psychiatric consultation when they evaluated the psychiatric symptoms of the HCW according to the DSM5, and looked for frequent themes of difficulties linking with the work expressed by HCW that emerged. Using content analysis methodology, two coders reviewed all interview scripts for recurrent themes, which they then categorized and sub-categorized, while comparing emerging categories to each other to determine their substance and significance [30]. A recurrent theme was defined as a theme occurring more than twice in the interviews of two different HCWs. For a theme occurrence to be retained, it had to be noted by the two coders in their qualitative analysis of the interview report. In the event of a coding discrepancy, a discussion between the coders and the rest of the research team took place, in order to conclude on the appropriate coding and improve inter-rater reliability. According to the triangulation method, results were shared with members of the research team who did not contribute to the qualitative analysis to check if the results looked coherent.

## 3. Results

### 3.1. CovidPsy Consultation Activity

#### 3.1.1. Quantitative Analysis of Clinical Data

We found that 38.89% of HCW had a psychiatric history (*n* = 7) and 83.33% reported entourage support (*n* = 15), which is considered as a protective factor (Table 1).

Average PDI scores were very high (µ = 28.14, [1–51]), with 11 (78.57%) having PDI scores ≥ 15, suggesting an important risk of future PTSD, similar to PDEQ scores (µ = 21.85; [9–39]), with 11 (78.57%) having scores ≥ 15 (Table 2). In addition, we found high HADS-anxiety scores (µ = 13.25; [6–20]), with 11 (68.75%) ≥ 11, suggesting pathological and severe anxiety. The mean HADS-depression scores were less high (µ = 8.31, [1–16]), with 5 (31.25%) having higher than 11. Burnout symptoms were very frequent: 4 (23.53%) with severe scores and 6 (35.29%) with moderate scores of emotional exhaustion (µ (EE) = 26.35, 7–46); 3 (17.65%) with moderate and 4 (23.53%) with severe depersonalization levels, (µ (PD) = 6.58; [1–22]); 2 (11.76%) with low and 6 (35.29%) with moderate personal achievement levels (µ (PA) = 31.70, 6–45); and 7 (38.88%) with a high level of burnout syndrome according to the MBI-HSS (Table 2, Figure 1) On the basis of our semi-structured interviews, we concluded that 7 (38.89%) had an ASD or a PTSD diagnosis, and 4 (22.22%) had an adjustment disorder according to DSM-5 criteria. Furthermore, 1 (5.55%) had a major depressive episode, and 1 (5.55%) had no psychiatric diagnosis, with some HCW having combined diagnoses (Figure 2). Only 7 (38.89%) HCW had to stop working for a period of time, and 5 (27.77%) were referred to a psychiatrist- psychotherapist or a psychologist psychotherapist follow-up, and 13 (72.22%) did not need to be referred because they were sufficiently clinically improved after the end of the CovidPsy consultation. In addition, 11(61.11%) had received an anxiolytic prescription, 7 (38.89%) had an antidepressant prescription, and 2 (11.11%) had no prescription of psychotropic medication (Table 1).

#### 3.1.2. Qualitative Analysis of Clinical Data: Difficulties and Stress Factors during the First Wave of the COVID-19 Pandemic

The two reviewers found a total of eleven themes that were the following: (1) lack of recognition (27.78%; *n* = 5); (2) conflict of values (11.11%; *n* = 2); (3) feeling of failure (11.11%; *n* = 2); (4) feeling of guilt (11.11%; *n* = 2); (5) feeling of insecurity (33.33%; *n* = 6); (6) feeling of abandonment (50%; *n* = 9); (7) fear of contamination (44.44%; *n* = 7); (8) multiple patients deaths (50%; *n* = 9); (9) difficulties with service change (33.33%; *n* = 6); (10) isolation for (33.33%; *n* = 6); (11) no perceived support by the hierarchy (15.16%; *n* = 3) (Table 3).

The reviewers gathered five themes in the first category of global themes that we called” traumatic stressors”, which correspond to exposure to multiple deaths, fear of contamination, feeling insecure, and feeling guilty. Hospital workers mostly expressed difficulties in coping with uncertainty related to COVID-19, and especially coping with contradictory information about how contamination occurred, and the lack of knowledge about the infection itself. In general, HCW suffered from having to adapt many changes, which created feelings of insecurity. We found that 50% (*n* = 9) expressed a suffering in line with their direct exposure to the multiple deaths, and that constitutes a traumatic event according to ASD and PTSD criteria in DSM5. We found that 44.44% (*n* = 7) were afraid of being contaminated and subsequently infecting family members or their social circle. This fear can be relayed to the fear to reexperience what they lived in a traumatic way with their patients, and this criterion of fear belongs to ASD criteria and PTSD criteria diagnosis such as in the question concerning PCL5. We found that 33.33% (*n* = 6) felt insecure, and this feeling can be relayed with the anxiety that is assessed in HADS and is also included in the intrusive symptoms of ASD and PTSD. We found that 11.11% (*n* = 2) felt guilty, knowing that there is a specific question in PCL5 concerning the fact of blaming oneself (10th question in PCL5).

The reviewers gathered seven themes in a second global category of themes that we called “work-related stress” which appeared to align with work stressors corresponding to change of position imposed (44.44%; *n* = 8), lack of recognition (27.78%; *n* = 5), and feelings of abandonment by their hierarchical superiors (50%, *n* = 9), feelings of incompetence (22.22%; *n* = 4), conflict of values (11.11%; *n* = 2), feelings of failure (11.11%; *n* = 2), and isolation (33.33%; *n* = 6) (Table 3).

We performed *t*-tests to compare clinical scores for participants who reported one of the “work-related stress” themes with the others and found that the difference was significant for the themes called “feelings of incompetence” (*p* = 0.02; [3.07; 29.10]) and “conflict of values” (*p* = 0.02, [4.08; 37.05]), concerning MBI-EE scores (Table 4). The ”conflict of values” theme looked associated with MBI-DP scores (0.0009; [−23.94; −7.59]), and so did the “lack of recognizing” theme (0.001; [−30.57; −9.10]) (Table 4).

## 4. Discussion

The high majority of the HCW who came to us worked in COVID-19 units and were either physicians or nurses, confirming what others have found regarding risk factors of stress related to working in COVID-19 units during this crisis [4,5].

The burnout and traumatic stress-related disorders (ASD or PTSD), such as anxiety and depressive symptoms, were found in high proportions for the HCW that consulted the permanence (Figure 1 and Figure 2). Most individuals suffered from working conditions related to their own safety, even if they themselves were not considered at risk to develop a severe form of COVID-19. For certain HCW, they did not feel enough supported by their colleagues, and/or hierarchy. In previous studies, the high prevalence of PTSD has been confirmed, since certain variables were found to be of particular relevance as risk factors as well as resilience factors, including exposure level, working role, years of work experience, social and work support, job organization, quarantine, age, gender, marital status, and coping styles [6,31]. Fear of contamination concerned 44.44% (*n* = 7) of the HCW who consulted, similar to an Italian study based on an online questionnaire that concluded a higher risk perception, level of worry, and knowledge as related to COVID-19 infection compared to the general population [32].

In our study, participants presented burnout symptoms: 35.29% presented a moderate and 23.53% a severe emotional exhaustion level; 17.65% a moderate and 23.53% a severe depersonalization level; and 11.76% a low and 35.29% a moderate personal achievement (Figure 1). These results confirm those of an Italian study that showed five weeks after the beginning of the outbreak that almost 33% presented high scores of emotional exhaustion, and almost 25% reported high levels of depersonalization [33], with a meta-analysis that identified a burnout prevalence of 37.4% [5]. By analyzing the *t*-tests results for themes, we can propose some hypotheses concerning the mechanisms underlying the psychiatric issues for HCW. The fact that we found an association between the feeling of incompetence and MBI-EE score was not surprising knowing the questions included in this subscore that concern incompetence. The influence of the presence of conflict of values in HCW and the lack of recognizing on the MBI-DP scores confirmed known data concerning risk factors of burnout [8] (Table 3).

To the best of our knowledge, there is no previous qualitative study concerning physical psychiatric consultations with HCW during the first wave of COVID-19, with most interventions consisting of hotlines [34,35], and not face-to-face consultations which offer more in-depth care.

Although we did not obtain follow-up data, we noted only a few participants needed to be referred for psychiatric or psychological follow-ups at the end of the CovidPsy care, which was sufficient for the majority of the HCW who consulted. Indeed, 72.22% of the HCW were sufficiently clinically improved after the end of the CovidPsy consultation and we could stop the CovidPsy consultation without referral to a psychiatrist or psychologist, suggesting that the intervention was early and in a preventive process of psychiatric issues.

This suggests that the efficiency of early detection and care of HCW with psychological suffering to reduce long term health and work consequences would need to be confirmed in a prospective study design. The work-related stress linked with work overload, lack of recognition, and feelings of abandonment by the hierarchy, suggest certain management principles at hospitals, such as reinforcement of staff during a crisis, supporting the efforts of HCW, and accompanying them, are necessary. Some authors suggested a theoretical model of emotional contagion that was observed in other groups during pandemics, which could explain our results regarding the psychiatric issues in the group of HCW [36,37].

There are several limitations in our study. First, this description of a consultation activity and the psychiatric screening during the first wave cannot be generalized because of its small and sample size and its auto-selected characteristics. We chose to include only the HCW who decided by themselves to come at the permanence and who agreed to participate to the study, and so the sample was limited. There is a necessary recruitment bias because only the HCW who considered they needed support were included, although other HCW in fact needed it but did not come, and reciprocally, perhaps some HCW came although they did not require it.

Second, HCW who came to the psychiatric consultation constituted a small percentage of overall HCW (25 on 13′557 HCW). The psychiatric consultation was only one of several strategies implemented during the first wave in terms of the psychological support. Therefore, the first hypothesis to explain the low number of consultations is that HCW felt helped by other implemented strategies (e.g., psychological support provided close to care units, hypnosis sessions, and hotlines). The presence of psychologists within the COVID-19 units, who could be solicited for speaking in one-to-one settings, or for group interventions, likely played an extremely important role. These interventions were notably different from those coming from CovidPsy consultation. One could also hypothesize that shame and fear of judgement, due to the stigma of psychiatry, could explain this small number of consultations. Workers expressed that they were reluctant to benefit from a psychiatric consultation within the hospital during working hours as they considered that when they were present, their concentration should be on providing care, suggesting that the hospital setting did not facilitate the use of the permanence. Finally, we can argue that in the heart of the first wave, the vast majority of HCW did not feel the need to ask for help because they were motivated by their goal and their pride to accomplish their mission. During the post-crisis phases and successive COVID-19 waves, one can imagine that psychological support could still have been useful and perhaps more needed with time. The first reason may be a delay in psychic distress from the event, by virtue of an afterthought effect. The second reason relates to possible exhaustion over time with these additional COVID-19 waves and the absence of any possibility to recover between them. Therefore, we will have to be vigilant about potential long term psychological effects, particularly if we consider the prospective disillusionment and reconstruction phases of human services workers following a disaster [38].

Furthermore, we analyzed semi-structured interviews notes that were not exact verbatim records of the consultations, having been transcribed and interpreted by the psychiatrist in a clinical context. We did not record the interviews because it was a retrospective study about clinical notes obtained during semi-structured interviews. In fact, it is not generally acceptable to record clinical interviews in clinical practice. We recall that we did not have a research goal at the origin. Without systematic available verbatims, but just interview notes of the HCW written by the interviewer, we recognize that there was probably a bias of interpretation. We should haven recorded the interviews of the HCW in an originally scientific goal. Finally, we did not explore protective factors that could have been helpful for HCW during this unprecedented crisis, except in regards to entourage support, and we do not have long-term follow-ups to assess the evolution of the intervention. It limits the understanding of the mechanisms of psychiatric complications for HCW in the context of COVID-19 pandemics. In a prospective study, we should have anticipated this need for a larger assessment of intrinsic and extrinsic protective factors.

The experience of this consultation activity should help us in the future in the case of other epidemic waves or health crises. Generally, the difficulties that we encountered for the implementation of the CovidPsy consultation was in line with the emergency context and having to take very quick decisions to start the consultation. It is very important afterwards to be able to learn from this experience of creating permanence in emergency now that the health situation has calmed down.

Considering the low rate of consultation at the psychiatric permanence, this consultation seemed to have been organized and proposed too early, and should have been more useful later. In the future, this kind of consultation should be maintained for longer and well after the peak of hospitalizations. In all likelihood, this type of consultation would have been useful following the first wave of COVID-19, however we could not extend it due to the resumption of the usual clinical activities of psychiatrists and clinical specialist nurses in psychiatry of the psychiatric consultation. Indeed, while we were already deploring four successive waves of COVID-19 in the fall of 2021, the direction of our hospital communicated about a high rate of absenteeism. The hypothesis of delayed psychic consequences can be legitimately put forward. For the next important epidemic waves or similar health crisis, one will have to anticipate the necessity of providing psychiatric care for HCW and to find an organization that will be compatible with the usual activities, for example the creation of persistent personal workplaces to engage with mental health specialists.

We showed that the qualitative analysis identified subjective information about the difficulties that caused distress, and this will be helpful in elaborating preventive strategies for hierarchy concerning the negative effects of lack of recognizing and support, such as those of service change.

The intervention of psychiatrists and nurses with colleagues in mental distress is not easy because it is not a common situation. Training sessions were the occasion to have common clinical references for the activity. Once per week, a coordination meeting was held with the people in charge of the support systems. Moreover, several training sessions were provided to the team, recalling an important theoretical-clinical basis for colleagues who were less familiar with this clinical field and to transmit guidelines for the permanence, from the reception of the healthcare worker to the end of the care. We additionally provided sessions on advice for preventing exhaustion and vicarious trauma for the psychiatric consultation team who were exposed to heavy emotional burden arising from caring for their colleagues, as well as to the global effect of the pandemic. Daily team meetings sessions were organized to analyze and discuss the clinical situations and their care. In the future, attention should be paid to potential psychic complications of the psychiatric team, and prevention tools like those we used should be implemented like team exchanges, and prevention sessions about vicarious trauma. Moreover, for all HCW, sessions on the prevention of psychiatric issues in the workplace should be organized throughout training and regularly throughout professional life to reduce these risks in the case of other epidemic waves or health crises.

## 5. Conclusions

This psychiatric consultation for HCW experience provides confirmation of the psychiatric consequences during the first wave of COVID-19, and the type of responses to prevent and early treat potential psychiatric complications. ASD, PTSD, burnout and anxiety symptoms were the most frequent psychiatric outcomes observed. Long-term and psychiatric consequences on mental health are expected in HCW that worked during the first wave of COVID-19. A psychiatric permanence for HCW allowed early intervention to prevent and treat psychiatric issues in the context of COVID-19 pandemics. Further studies would be needed to assess the efficiency of this kind of intervention for HCW. Considering the risk of delayed psychiatric issues, the need of intervention should not be limited in the time and should be offered to HCW even after the crisis period.

## Figures and Tables

**Figure 1 ijerph-19-04780-f001:**
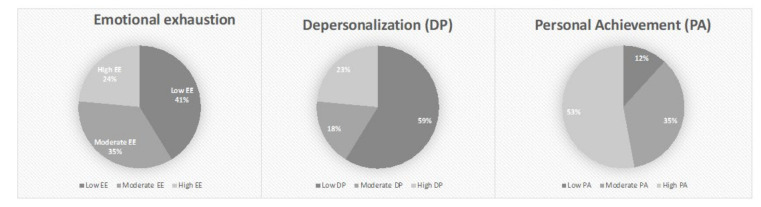
Three dimensions of Maslach Burnout Inventory-Human Services Survey (MBI-HSS) scores distribution.

**Figure 2 ijerph-19-04780-f002:**
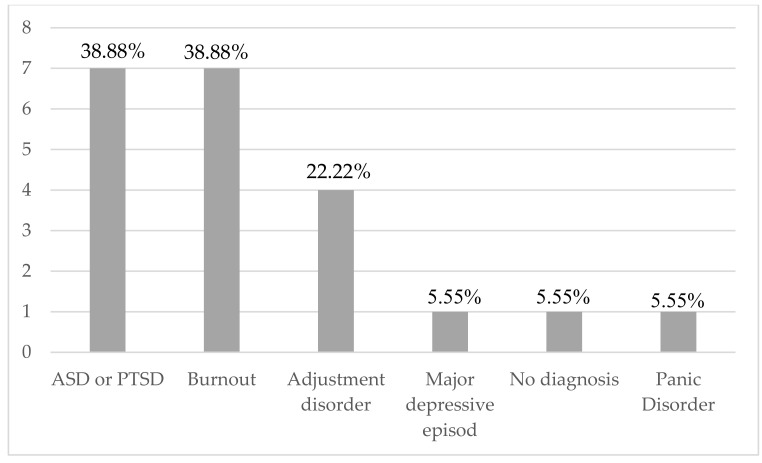
Psychiatric diagnosis retained according to DSM-5 criteria diagnosis and MBI-HSS. ASD: Acute Stress Disorder; PTSD: Post-Traumatic Stress Disorder.

**Table 1 ijerph-19-04780-t001:** Socio-demographic characteristics, clinical data and characteristics of the follow-up.

	Total (*N* = 18); N(%)
Socio-Demographic Characteristics	
Age (mean, min-max)	40,67; [25–58]
Gender	
- Female	14 (77.78)
- Male	4 (22.22)
Profession	
- Physician	2 (11.11)
- Nurse	9 (50)
- Medical student	2 (11.11)
- Other healthcare worker	4 (22.22)
- Administrative hospital staff	1 (5.55)
Marital status	
- Single	3 (16.67)
- Married or living as a couple	15 (83.33)
Entourage support	
- Yes	15 (83.33)
- No	3 (16.67)
Working in COVID-19 ward	
- Yes	13 (72.22)
- No	5 (27.78)
Clinical data	
Medical history	
- Yes	8 (44.44)
- No	7 (55.56)
Psychiatric history	
- Yes	7 (38.89)
- No	11 (61.11)
- No information	
Current psychotropic medication	
- Yes	2 (11.11)
- No	16 (88.89)
Trauma history	
- Yes	9 (50)
- No	9 (50)
SARS-CoV-2 contamination	
- Yes	4 (22.22)
- No	14 (77.78)
Medical risk factors for severe COVID-19 form *	
- Yes	2 (11.11)
- No	16 (88.89)
Number of sessions by healthcare worker (HCW)	µ = 2.68; [−1; 11]
Characteristics of the follow-up	
Duration of consultation	µ = 89 min (45–150 min)
Referral to a psychiatrist or a psychologist follow-up	
- Yes	5 (27.78)
- No	13 (72.22)
Sick leave	
- Yes	7 (38.89)
- No	11 (61.11)
Prescription of a psychotropic medication in the context of CovidPsy consultation	
- Anxiolytic medication	11 (61.11)
- Antidepressant medication	2 (11.11)
- No psychotropic prescription	7 (38.89)

* diabetes, obesity, hypertension, history of heart failure, ischaemic heart disease, solid organ tumours, chronic obstructive pulmonary disease (COPD), chronic respiratory disease, chronic.

**Table 2 ijerph-19-04780-t002:** Results of Peritraumatic Dissociative Experiences Questionnaire (PDEQ), Peritraumatic distresses inventory (PDI), Posttraumatic Stress Disorder Checklist for DSM-5 (PCL-5), the Hospital Anxiety and Depression scale (HADS) and Maslach Burnout Inventory-Human Services Survey (MBI-SS) scales.

PDEQ (*n* = 14)	
PDEQ score (mean, min-max)	21.85 (9–39)
- Positive PDEQ (N, %))	11 (78.57)
- PDI (*n* = 14)	
PDI score (mean, min-max)	28.14 (1–51)
- Positive PDI (N,%)	11 (78.57)
- PCL-5 (*n* = 4)	
Positive PCL-5 (N, %)	4 (100)
- PCL-5 (mean, min-max)	30 (33–47)
- HADS-A (*n* = 16)	
Positive HADS-A (N, %)	11 (68.75)
- HADS-A (mean, min-max)	13.25 (6–20)
- HADS-D (*n* = 16)	
Positive HADS-D (N, %)	5 (31.25)
- HADS-D (mean, min-max)	8.31 (1–16)
- MBI-HSS (*n* = 17)	
MBI-EE (N, %)	
- Low level (sub-score ≤ 21) (N, %)	7 (41.8)
◦ Moderate (21 < sub-score ≤ 32)	6 (35.29)
◦ High level (sub-score > 32)	4 (23.53)
◦ MBI-DP (N, %)	
- Low level (sub-score ≤ 6)	10 (58.82)
◦ Moderate (21 < sub-score ≤ 12)	3 (17.65)
◦ High level (>12)	4 (23.53)
◦ MBI-PA (N, %)	
- Low level (sub-score ≤ 22)	2 (11.76)
◦ Moderate (22 < sub-score ≤ 32)	6 (35.29)
◦ High level (sub-score > 32)	9 (52.94)

**Table 3 ijerph-19-04780-t003:** Qualitative analysis of the 47 written interview notes with HCW.

Perceived Difficulties at Work in Link with COVID-19 Outbreak (*N* = 18)	N(%)
Traumatic stressors	
- Fear of contamination at work	7 (38.89)
- Feeling insecure	6 (33.33)
- Multiple deaths of patients	9 (50)
- Feeling of guilty	2 (11.11)
Work related stress	
- Work change imposed by the COVID-19 context	8 (44.44)
- Lack of recognition	5 (27.78)
- Feeling abandoned by hierarchy	9 (50)
- Feeling of incompetence	4 (22.22)
- Conflict of values	2 (11.11)
- Feeling of failure	2 (11.11)
- Isolation	6 (33.33)
- No perceived support by the hierarchy	3 (16.67)

**Table 4 ijerph-19-04780-t004:** Student tests for work-related stress themes.

	MBI−EE	MBI−DP
Lack of recognitiong	0.94; [−17.26; 16.26]	0.001 **; [−30.57; −9.10]
Conflict of values	0.02 *; [4.08; 37.05]	0.0009 ***; [−23.94; −7.59]
Feelings of incompetence	0.02 *; [3.07; 29.10]	0.25; [−14.13; 3.97]
Feelings of abandonment	0.62; [−10.58; 17.02]	0.96; [−7.53; 7.86]
Difficulties with service change	0.19; [−5.55; 25.51]	1; [−12.18; 12.18]
Feelings of isolation	0.94; [−14.6372; 13.6372]	0.62; [−10.36; 6.36]
No perceived support by hierarchy	0.07; [−28.89; 1.32]	0.62; [−10.36; 6.36]

* *p* > 0.05; ** *p* > 0.005; *** *p* > 0.0005.

## Data Availability

Not applicable.

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
