# Peer review of "Implementation of a Psychiatric Consultation for Healthcare Workers during First Wave of COVID-19 Outbreak"

_ijerph, 2022, doi:10.3390/ijerph19084780_

Round 1

Reviewer 1 Report

Although the aim of the paper is interesting, as it deals first and foremost with psychological problems occurring in health professionals, there are several problems that detract from the quality of the article:

1. What is HCW? It is not specified anywhere.

2. It is not explained in depth how they identified the risks for health workers. They say that they rely on previous epidemics, but they could have done a general assessment of the psychological state and needs of the workers in their hospital. It should be remembered that the impact of COVID has not been the same in all countries, nor has the hospital pressure and consequently the psychological impact on healthcare workers. 

3. They comment that the interventions consisted of three sessions, except for some cases. This is an issue that should be explored further. Why only three sessions, on what basis, and what exceptions?

4. Qualitative analysis of semi-structured interviews is carried out, but it does not specify what these interviews are like. This is an issue that needs to be described.

5. Regarding the interventions, what happened if people still showed symptomatology after three sessions? What was done with these people? What help was given to them? Why was there no evaluation after the intervention?

Reviewer 2 Report

Dear Editor,

Thank you for your invitation to review this manuscript.

General comments: I do not believe that the design of the study can answer the aims stated by the authors.

  • The small number of cases n=18 offers little basis for establishing reliability or to generalize the findings to a wider population of people, and so, it is not possible to identify the potential psychiatric disorders in healthcare workers (HCW).

  • The authors have described the implementation of a mental health consultation for HCW, but only 25 of 13557 HCW attended. This is a major flaw since it is impossible to know how can 25 HCW represent the total population of HCWs. For example, only 2 physicians were included vs 2 medical students, did a medical student had the same stress induced work as a physician during the first wave of the COVID-19 pandemic?

  • The authors also used a qualitative approach to analyze 47 clinical written notes regarding mental health of HCW. This was an indirect approach since the authors of the paper did not directly interview the patients and the risk of interpreting someone else’s interpretation is great. How was the quality of the clinical records measured? Again, 47 consultations is a very small number.

I am sorry to disappoint the authors but I cannot advise publication.

Reviewer 3 Report

I have reviewed the manuscript entitled "Implementation of a Psychiatric Consultation for Healthcare 2 Workers During First Wave of COVID-19 Outbreak." The manuscript is relevant for the audience of this journal and explores the psychological suffering (anxiety, depression, insomnia, etc) of the healthcare 2 workers during the covid-19 pandemic. 

To be published I suggest that authors give more context about the importance of emotions (negative emotions) during the covid-19 pandemic, introducing the concept of emotional contagion that sociologists and psychologists have observed in other groups during the pandemic.

For this I suggest revising these papers to enrich the first part of the manuscript and elaborating a more detailed theoretical frame: 

Belli, S., & Alonso, C. (2021). COVID-19 pandemic and emotional contagion. Digithum, (27), 1-9.

Rebughini, P. (2021). A sociology of anxiety: Western modern legacy and the Covid-19 outbreak. International Sociology36(4), 554-568.

Round 2

Reviewer 2 Report

The authors have now stated more clearly the limitations of the study. Nonetheless, I still do not believe that the design of the study can answer the aims stated by the authors.

Author Response

Dear Reviewer,

We thank you again for your attentive lecture of our manuscript and for your comments.

Lamyae Benzakour, on the behalf of all coauthors.

Reviewer 3 Report

I accept the manuscript.

Author Response

(The authors gave the same response as above.)
